# Quantification of Plasma and Urine Thymidine and 2’-Deoxyuridine by LC-MS/MS for the Pharmacodynamic Evaluation of Erythrocyte Encapsulated Thymidine Phosphorylase in Patients with Mitochondrial Neurogastrointestinal Encephalomyopathy

**DOI:** 10.3390/jcm9030788

**Published:** 2020-03-13

**Authors:** Karin Kipper, Max Hecht, Natalicia J. Antunes, Lynette D. Fairbanks, Michelle Levene, Sema Kalkan Uçar, Andrew Schaefer, Emma L. Blakely, Bridget E. Bax

**Affiliations:** 1Analytical Services International Ltd., St George’s University of London, Cranmer Terrace, London SW17 0RE, UK; karin.kipper@gmail.com (K.K.); hecht@ut.ee (M.H.); nataliciaja@gmail.com (N.J.A.); 2University of Tartu, Institute of Chemistry, 14a Ravila Street, 50411 Tartu, Estonia; 3Department of Pharmacology, Faculty of Medical Sciences, State University of Campinas (UNICAMP), Campinas, SP 13083-881, Brazil; 4Clinical Pharmacology, William Harvey Research Institute, Barts and The London School of Medicine and Dentistry, Queen Mary University of London, London EC1M 6BQ, UK; 5The Purine Research Laboratory, St Thomas’ Hospital, London SE1 7EH, UK; lynette.fairbanks@viapath.co.uk; 6Molecular and Clinical Sciences, St George’s University of London, London SW17 0RE, UK; mlevene@sgul.ac.uk; 7Division of Inborn Error of Metabolism, Ege University Medical Faculty, 35100 Izmir, Turkey; sema.kalkan.ucar@ege.edu.tr; 8The NHS Highly Specialised Service for Rare Mitochondrial Disorders, Newcastle upon Tyne NE2 4HH, UK; a.schaefer@nhs.net (A.S.); emma.watson33@nhs.net (E.L.B.)

**Keywords:** LC–MS/MS, method validation, thymidine, 2’-deoxyuridine, mitochondrial neurogastrointestinal encephalomyopathy, erythrocyte encapsulated thymidine phosphorylase, EE-TP

## Abstract

Mitochondrial neurogastrointestinal encephalomyopathy (MNGIE) is an ultra-rare disorder caused by mutations in *TYMP*, leading to a deficiency in thymidine phosphorylase and a subsequent systemic accumulation of thymidine and 2’-deoxyuridine. Erythrocyte-encapsulated thymidine phosphorylase (EE-TP) is under clinical development as an enzyme replacement therapy for MNGIE. Bioanalytical methods were developed according to regulatory guidelines for the quantification of thymidine and 2’-deoxyuridine in plasma and urine using liquid chromatography-tandem mass spectrometry (LC–MS/MS) for supporting the pharmacodynamic evaluation of EE-TP. Samples were deproteinized with 5% perchloric acid (v/v) and the supernatants analyzed using a Hypercarb column (30 × 2.1 mm, 3 µm), with mobile phases of 0.1% formic acid in methanol and 0.1% formic acid in deionized water. Detection was conducted using an ion-spray interface running in positive mode. Isotopically labelled thymidine and 2’-deoxyuridine were used as internal standards. Calibration curves for both metabolites showed linearity (*r* > 0.99) in the concentration ranges of 10–10,000 ng/mL for plasma, and 1–50 µg/mL for urine, with method analytical performances within the acceptable criteria for quality control samples. The plasma method was successfully applied to the diagnosis of two patients with MNGIE and the quantification of plasma metabolites in three patients treated with EE-TP.

## 1. Introduction

Mitochondrial neurogastrointestinal encephalomyopathy (MNGIE) is a fatal autosomal recessive disorder caused by mutations in *TYMP*, the gene that encodes for thymidine phosphorylase (EC 2.4.2.4). The resulting enzyme deficiency leads to elevated concentrations of the deoxyribonucleosides, thymidine and 2’-deoxyuridine in cellular and extra-cellular compartments, which then generate imbalances within the mitochondrial deoxyribonucleotide pools, causing mitochondrial DNA (mtDNA) mutations and depletion, and ultimately mitochondrial dysfunction [1]. Currently there are no licensed therapies available for patients with MNGIE, although a number of experimental therapeutic approaches have been investigated. The therapeutic approaches have the common aim of normalizing or reducing the systemic accumulations of the deoxyribonucleosides, and thereby prevent further mitochondrial damage and slow disease progression [2,3,4,5,6,7,8,9,10,11,12,13]. We have regulatory approval in the United Kingdom for a trial of erythrocyte-encapsulated thymidine phosphorylase (EE-TP), an enzyme replacement therapy for the treatment of MNGIE. One of the trial’s primary objectives is the longitudinal assessment of the pharmacodynamic effects of EE-TP for optimizing dose level selection and establishing a therapeutic window for treatment. These pharmacodynamic assessments include the measurement of plasma and urine thymidine and 2’-deoxyuridine concentrations [14]. There is thus a requirement for validated bioanalytical methods in accordance with the regulatory guidelines for the quantification of thymidine and 2’-deoxyuridine in collected plasma and urine samples. In accordance with the European Medicines Agency (EMA) guidelines, validation parameters that should be considered include selectivity, accuracy, precision, calibration curve, sensitivity, stability, matrix effects and recovery [15]. Several methods, including high-performance liquid chromatography with UV detection (HPLC-UV) and high-performance liquid chromatographic–tandem mass spectrometry analysis (LC–MS/MS) have been developed for thymidine quantification in plasma and urine, however these methods lack the fundamental attributes required for analytical validation [6,16,17]. More recently Mohamed et al. reported a validated HPLC-UV assay for the simultaneous measurement of thymidine and 2’-deoxyuridine in the plasma of patients with MNGIE, where the lower limit of quantification was 0.5 µg/mL (~2 µmol/L) for both deoxyribonucleosides [18]. However, plasma concentrations of thymidine and 2’-deoxyuridine in unaffected individuals are undetectable, being below the commonly reported limit of 0.05 µmol/L and beyond the sensitivity limit of HPLC–UV detection [2]. In addition, patients with MNGIE with the late onset disease phenotype have be reported with plasma metabolite concentrations in the range of 0.05–4 µmol/L for thymidine and 0.05–5 µmol/L for 2’-deoxyuridine, providing the opportunity for false negative diagnoses of MNGIE using currently available methods [19]. Li et al. developed and validated a sensitive LC–MS/MS assay to evaluate the therapeutic response of capecitabine in patients with advanced colorectal cancer, but this was for the measurement of plasma 2’-deoxyuridine only [20]. We report here the first validated LC–MS/MS assay for the simultaneous quantitation of thymidine and 2’-deoxyuridine in plasma with a lower limit of detection within the range of healthy individuals. We also report the first validated LC–MS/MS method for the quantification of thymidine and 2’-deoxyuridine in urine.

## 2. Materials and Methods

### 2.1. Reagents and Materials

Thymidine, 2’-deoxyuridine and formic acid (LC-MS grade) were obtained from Sigma Aldrich (Poole, UK). Internal standards (IS) thymidine-1’,2’,3’,4’,5’-^13^C_5_ and 2’-deoxyuridine-1’,2’,3’,4’,5’-^13^C_5_ were from Toronto Research Chemicals (Toronto ON, Canada) and 70% perchloric acid (PCA) was obtained from VWR (Lutterworth, UK). Methanol (HPLC grade) was supplied by Rathburn (Walkerburn, Scotland). Deionized water was prepared in-house (<1.0 µS/cm). Analyte-free dialyzed human plasma (with K_2_EDTA) was obtained from TCS Biosciences Ltd (Buckingham, UK). Analyte-free human plasma containing K_3_EDTA, lithium heparin, analyte-free human serum from individual donors and whole blood (for the preparation of hemolyzed plasma) were obtained from Biological Speciality Corporation (Colmar, PA, USA). Hyperlipidaemic plasma containing 247 mg/dL cholesterol was supplied by Seralab (Haywards Heath, UK). Urine was obtained from in-house healthy volunteers. 

### 2.2. Preparation of Calibrators, Quality Control (QC), and IS Solutions

#### 2.2.1. Plasma Assay

Calibrator and Quality Control (QC) stock solutions containing 1 mg/mL of either thymidine or 2’-deoxyuridine were prepared in methanol and stored at −18 to −20 °C. Two sub-stock solutions containing a combination of thymidine and 2’-deoxyuridine were prepared from each Calibrator and QC stock solution containing 10 µg/mL and 0.1 µg/mL of each analyte in analyte-free dialyzed human plasma. Calibrator standards were prepared by appropriate dilution of the Calibrator sub-stock solutions with analyte-free dialyzed human plasma to contain 10, 50, 100, 200, 500, 1000, 5000 and 10,000 ng/mL of thymidine and 2’-deoxyuridine (nominal concentrations). QC standards were similarly prepared by the appropriate dilution of the combined QC sub-stocks at concentrations 10 µg/mL and 0.1 µg/mL. QC standard concentrations in dialyzed human plasma were 10, 30, 500 and 7500 ng/mL of thymidine and 2’-deoxyuridine (nominal concentrations). Matrix-free QC standards for determining matrix effects were prepared by the formulation of two combined sub-stock solutions, 0.1 µg/mL and 10 µg/mL (both made up from the thymidine and 2’-deoxyuridine QC stock solutions and diluted in methanol), followed by a further dilution in methanol to contain analyte concentrations 30 and 7500 ng/mL, respectively. Once prepared, Calibrator and QC standards were aliquoted into 1.5 mL micro-centrifuge tubes and stored frozen at −18 to −20 °C until required for analysis. IS stock solutions for thymidine-1’,2’,3’,4’,5’-^13^C_5_ and 2’-deoxyuridine-1’,2’,3’,4’,5’-^13^C_5_ (approximately 1 mg/L) were prepared in methanol and stored at −18 to −20 °C. A working IS solution containing a combination of thymidine and 2’-deoxyuridine (each of an approximate concentration of 500 ng/mL) was prepared by diluting 0.1 mL of each IS in 200 mL of methanol. The working IS solution was stored at −18 °C to −20 °C. 

#### 2.2.2. Urine Assay

Calibrator and QC stock solutions containing 1 mg/mL of either thymidine or 2’-deoxyuridine were prepared in methanol and stored at −18 to −20 °C. A sub-stock solution containing a combination of thymidine and 2’-deoxyuridine was prepared from each Calibrator and QC stock solution containing 50 µg/mL of each analyte in analyte-free human urine (containing 5% of PCA). Calibrator standards were prepared by appropriate dilution of the calibrator sub-stock solution with analyte-free human urine (containing 5% of PCA) to contain 1, 5, 7.5, 10, 25 and 50 µg/mL of thymidine and 2’-deoxyuridine. QC standards were similarly prepared by the appropriate dilution of the combined QC sub-stock with human urine (containing 5% PCA) to contain 1, 3, 15 and 37.5 µg/mL of thymidine and 2’-deoxyuridine. Matrix-free QC standards for determining matrix effects were prepared by the formulation of the sub-stock solution 50 µg/mL (made up from the thymidine and 2’-deoxyuridine QC stock solutions and diluted in methanol), followed by a further dilution in methanol to contain analyte concentrations of 3 and 37.5 µg/mL. Once prepared, Calibrator and QC standards were aliquoted into 1.5 mL micro-centrifuge tubes and stored frozen at −18 to −20 °C until required for analysis. For stability studies a set of Calibrator and QC standards were prepared in human urine without PCA following the same dilution scheme as described above.

A combined working IS solution (containing thymidine and 2’-deoxyuridine each at an approximate concentration of 5 µg/mL) was prepared by diluting 1 mL of each IS (prepared as described in Section 2.2.1) in 200 mL of methanol. The working IS solution was stored at −18 °C to −20 °C. 

### 2.3. Sample Preparation

Plasma samples were prepared by adding 100 μL of working IS solution to 100 µL of each plasma Calibrator standard (except double blank, i.e. plasma sample containing no analyte or IS), QC standard and patient sample, followed by the addition of 100 μL of 7% PCA in deionized water. Urine samples were prepared by adding 50 μL of working IS solution to 50 µL of each Calibrator (except double blank, i.e., urine sample containing no analyte or IS) and QC standard, followed by the addition of 125 μL of 7% PCA in deionized water.

The samples were mixed for 5 minutes on a flatbed mixer and centrifuged at 13,000 rpm for 5 min. The supernatant (approximately 250 µL) was transferred into a clean 96-well plate or clean polypropylene autosampler vial and submitted for analysis by LC–MS/MS. For the analytical extraction recovery experiments, post extraction spiked plasma and urine samples were prepared by spiking analyte-free plasma with analytes and IS after extraction with PCA.

### 2.4. LC-MS/MS Instrumentation and Analytical Conditions

A 1290 binary LC pump was used with a 1290 Infinity IITM autosampler (Agilent Technologies, Santa Clara, CA, USA). The separation of analytes was performed using a Hypercarb column (30 × 2.1 mm, 3 µm, Thermo Fisher Scientific, Waltham, MA, USA) held at 60 °C in a column oven (MCT 1290, Agilent Technologies). The mobile phase consisted of 0.1 % formic acid in methanol (mobile phase B) and 0.1% formic acid in deionized water (mobile phase A) at a flow rate of 0.6 mL/min. The applied gradient program increased the % B content from 30% to 100% in 3.5 min, thereafter % B was kept at 100% for 0.5 min, and decreased to 30% in 0.2 min. The mobile phase B was then kept at 30% for 0.8 min. The total run time was 5 min. The injection volume was 10 µL. Analyte and IS detection was carried out by tandem mass spectrometry (MS) on an API4000 (AB Sciex, Concord, ON, Canada) with Positive Heated ion spray (Positive TurboIonSpray, MH+): m/z 228.997 → m/z 113.1 (quantifier ion, Q) and m/z 228.997 → m/z 117.1 for 2’-deoxyuridine; m/z 242.958 → m/z 126.9 (Q) and m/z 242.958 → m/z 117.0 for thymidine; m/z 234.007 → m/z 113.0 for 2’-deoxyuridine-1’,2’,3’,4’,5’-^13^C_5_, m/z 248.028 → m/z 127.0 for thymidine-1’,2’,3’,4’,5’-^13^C_5_. The following MS settings were used: probe voltage 5000 V, source temperature 400 °C, curtain gas, nebulizer gas and heater gas were set at 30 psig, optimised collision energies for compounds ranged from 13 to 15 V and the dwell times were set to 30 ms.

### 2.5. Method Validation

The LC–MS/MS methods were validated according to the EMA guidelines with respect to linearity, lower limit of quantitation (LLOQ), selectivity, inter- and intra-day precisions and accuracy, carry-over, matrix effect (matrix factor), recovery and stability [15].

Method linearity was evaluated by constructing calibration curves from eight plasma Calibrator standards in the range of 10 to 10,000 ng/mL and six urine Calibrator standards in the range of 1 to 50 µg/mL. The Calibrator sets consisted of six and eight different batches of each Calibrator standard (prepared in duplicate), respectively for plasma and urine, a blank (containing IS) and double blank (blank matrix without IS). The peak area ratios (analyte area: IS area) were calculated from the peak area data and plotted against the peak area ratio of the spiked concentration of the Calibrator standards. A weighted (1/*x*^2^) linear regression analysis was conducted to determine the slope of the calibration lines, intercept, and correlation coefficient (*r*) for the determination of the linearity of the method. The acceptance criteria for r was a value of 0.98 or greater. The lower limit of quantification (LLOQ) was determined as the lowest analyte concentration on the calibration curve that could be measured with a precision within ±20%, as indicated by the coefficient of variation (CV%), and accuracy within 80%–120%. Accuracy, which determines the proximity between the obtained experimental results and the predicted results, was calculated as follows: [(measured Calibrator concentration/actual spiked Calibrator concentration) × 100].

For the evaluation of selectivity, six blank human urine samples, nine blank plasma samples (dialyzed K_2_EDTA plasma and plasma containing K_3_EDTA, lithium heparin, cholesterol, 2.5%, 5%, 7.5% and 10% lysed whole blood) and two serum samples from different donors were individually analyzed and evaluated for interference using the proposed extraction procedure and chromatographic/mass spectrometric conditions, and compared with the responses obtained with calibrators containing the analytes at concentrations near to the LLOQ. The absence of interfering components will be accepted where the response from interfering compounds is less than 20% of the LLOQ for the analyte and less than 5% for the IS.

Inter-assay precision and accuracy were evaluated by analyzing six replicates of each of the four QC standards (for urine and plasma), together with one independent calibration standard curve, on three different days. Intra-day accuracy and precision were determined by the analysis of six replicates of the four QC standards (for urine and plasma), together with a calibration standard curve on the same day. Inter-assay and intra-assay precisions were expressed as CV% and will be accepted at the level of ≤15%, except for the LLOQ which should be ≤20%. Inter-assay and intra-assay accuracies were expressed as the percent ratio between the measured concentrations and the spiked concentration for each QC and should be between 85% and 115% of the spiked concentration, except for the LLOQ which should be between 80% and 120% of the spiked concentration.

To assess carry-over, extracted double banks were injected immediately following the injection of the highest Calibrator standards. Carry-over was considered to be acceptable if the peak area analyte signal in the double blank after highest calibrator is less than 20% of LLOQ and less than 5% of IS peak area. 

The potential suppression/enhancement of thymidine and 2’-deoxyuridine ionization by matrix components during LC–MS/MS analysis was evaluated by measuring the matrix effects as described by Matuszewski et al. [21] or as matrix factor described by EMA guidelines [15]. In addition, the IS normalised matrix factor was evaluated according to the recommendations of the EMA guidelines on bioanalytical method validation [15]. For the plasma assay, the matrix effects and matrix factors were determined in four different plasma samples (dialysed K_2_EDTA plasma, K_3_EDTA, and lithium heparin) and two serum samples. Additionally, hyperlipidaemic plasma (with total cholesterol concentration of 247 mg/dL) and haemolysed plasma (containing 2.5, 5, 7.5 and 10% of whole blood) were tested. For the urine assay, these effects were determined for thymidine and 2’-deoxyuridine in six different urine samples. Peak area measurements were obtained from six replicates of post extracted plasma and urine QC samples spiked with thymidine and 2’-deoxyuridine and their IS at the same concentrations (30 ng/mL and 7500 ng/mL for plasma; 3 µg/mL and 37.5 µg/mL for urine) and compared to the peak area measurements obtained from neat QC standards of similar concentrations prepared in methanol (i.e., in the absence of matrix). The percent matrix effects (ME%) were calculated as follows: [(Mean peak area in presence of matrix/Mean peak area in the absence of matrix) × 100]. The IS normalised matrix factors were also evaluated for both analytes in all matrices, where the percent IS normalised matrix factors (MF%) were calculated by substituting the peak area with the peak response ratio (analyte: IS) in the above equation. The results for matrix effects and matrix factors were considered acceptable if the lot-to-lot matrix variation is within ± 15%.

The analytical extraction recovery was determined by comparing the mean peak area ratio (analyte: IS) response of two extracted QC plasma standards (30 and 7500 ng/mL) and two extracted QC urine standards (3 µg/mL and 37.5 µg/mL) with the mean peak area ratio response of plasma and urine samples spiked post extraction at similar concentrations to the QC samples in six replicates.

An assessment of thymidine and 2’-deoxyuridine stability was conducted by comparing the concentration of analytes in six replicates of low and high concentration plasma and urine QC standards over time, in different storage conditions. The stability of urine QC standards was assessed with and without the addition of 5% PCA (v/v).

Short term stability evaluations of analytes in plasma were conducted after subjecting QC standards to ambient (20.6 °C to 22.1 °C) and refrigeration (1.5 °C to 9.2 °C) temperatures for 24 h before processing and analysis. Stability evaluations of urine QC standards were made after keeping samples at ambient temperature (20.6 °C to 22.1 °C) for 24 h and refrigeration temperature (1.5 °C to 9.2 °C) over 14 days.

The effect of three freeze-thaw cycles on thymidine and 2’-deoxyuridine stability in plasma and urine was assessed. Aliquots at each of the two plasma and urine QC standards were stored at −20 ± 0.5 °C for at least 24 h before thawing at ambient temperature.

Processed sample stability was assessed after maintaining all four extracted plasma and urine QC standards in the autosampler vials or injection well-plate, at 4 °C for 24, 48 and 72 h. 

The stability of thymidine and 2’-deoxyuridine stock solutions was evaluated after storage at −20 °C for 2.5 months.

The evaluated stability QC standards were considered stable when the mean accuracy and precision (CV%) is within ± 15% and ≤5% of the actual concentrations, respectively.

### 2.6. Clinical Application of Validated Plasma Assay

The validated plasma assay was applied to the pharmacodynamic analysis of three patients treated with EE-TP under a compassionate treatment program and to the diagnosis of two patients where a diagnosis of MNGIE was suspected.

The compassionate treatment study was conducted in an open label manner with participants receiving EE-TP in accordance with the provisions of Schedule 1 of The Medicines for Human Use (Marketing Authorisations etc.) Regulations SI 1994/3144, where Schedule 1 provides an exemption from the need for a marketing authorisation for a relevant medicinal product, which is supplied on an individual patient basis to fulfil a “special need”. The study was approved by the National Research Ethics Service Committee and participants provided written informed consent before participating in any study procedures.

For the analysis of plasma thymidine and 2’-deoxyuridine, 3.5 mL blood samples were collected into K_2_EDTA tubes. The blood was centrifuged at 1100–1300 × *g* for 10 min and the plasma supernatant collected and stored at −80 °C until LC-MS/MS analysis using the validated method.

## 3. Results and Discussion

### 3.1. Optimization of LC–MS/MS Conditions 

A range of different stationary phases, including C18, biphenyl and Hypercarb^TM^, were tested to obtain appropriate retention and separation of analytes. Biphenyl and Hypercarb^TM^ columns both provided a good retention of highly hydrophilic analytes over the C18 stationary phase. The hypercarb^TM^ column, however, was selected based on a lower background noise, and thus providing an improved selectivity for analytes and the IS solutions. Retention times in plasma and urine were 2.0 minutes for thymidine and thymidine-1’,2’,3’,4’,5’-^13^C_5_ and 1.2 minutes for 2’-deoxyuridine and 2’-deoxyuridine-1’,2’,3’,4’,5’-^13^C_5_. 

### 3.2. Calibration Curve, LLOQ and Selectivity

Figure 1 show typical calibration curves for thymidine and 2’-deoxyuridine in plasma and urine. The calibration curves demonstrated good linearity over the analyte concentration ranges of 10 to 10,000 ng/mL in plasma and 1 to 50 µg/mL in urine. The back-calculated analyte concentrations from the calibration curves remained within ± 15% of accuracy; plasma thymidine—4.1 to 9.6%, plasma 2’-deoxyuridine—4.2 to 4.1%, urine thymidine—2.0 to 4.7%, and urine 2’-deoxyuridine—3.9 to 3.1%. The correlation coefficient, *r*, between concentration and peak area ratio for all thymidine and 2’-deoxyuridine curves were >0.99 in both plasma and urine. The coefficients of variation of the slopes of the calibration curves were 3.1% (plasma thymidine), 5.5% (plasma 2’-deoxyuridine), 4.2% (urine thymidine) and 4.7% (urine 2’-deoxyuridine). For the plasma assay, the LLOQ was set at 10 ng/mL, with between-assay accuracy of 95.7% and precision of 6.6% for thymidine, and with between-assay accuracy of 103.6% and precision of 4.4% for 2’-deoxyuridine. This is the most sensitive validated analytical method reported to date for thymidine and 2’-deoxyuridine, having a LLOQ within the range found in healthy, unaffected individuals. For the urine assay, the LLOQ was set at 1 µg/mL, with between-assay accuracy of 104.6% and precision of 11.5% for thymidine and with between-assay accuracy of 107.5% and precision of 8.2% for 2’-deoxyuridine. The accuracy and precision for LLOQ samples for both analytes and assays remained well within the set criteria of 80–120% and ± 20%, respectively. 

Figure 2 shows representative chromatograms of extracted double blank plasma (no analyte or IS), and plasma samples spiked with thymidine and 2’-deoxyuridine at the LLOQ (10 ng/mL) and upper limit of qualification (10,000 ng/mL). Figure 3 shows typical chromatograms of extracted double blank urine (no analyte or IS), and urine samples spiked with thymidine and 2’-deoxyuridine at the LLOQ (1 µg/mL) and upper limit of qualification (37 µg/mL). No interfering peaks from endogenous compounds were observed at the retention times of the analytes and the ISs. 

The selectivity for thymidine and 2’-deoxyuridine were 0% in all urine, plasma and serum samples tested, including plasma containing cholesterol and lysed whole blood, therefore demonstrating no interference in the assay response using the proposed extraction procedure and chromatographic/mass spectrometric conditions.

### 3.3. Accuracy and Precision 

Data for the intra- and inter-day accuracy and precision for the plasma and urine assays is presented in Table 1. Accuracies for both analytes were between 93.83% and 104.85% in plasma and 96.14% to 107.45% in urine, including the QC standard at the LLOQ. These values are within the set criteria of 85% to 115% (for the three highest QC standards). Assay precisions remained well below the set criteria of ± 15% for both plasma and urine assays, including the QC samples at the LLOQ level, therefore depicting a high precision of the methods. The fact that the precisions and accuracies of the QC standards at the level of the LLOQ were within the criteria range for the three higher QC standards indicate the possibility of lowering the LLOQ levels or required volume of matrix for the sample preparation while still remaining within the limits of permitted accuracy and precision.

### 3.4. Recovery and Matrix Effect

Thymidine recoveries in plasma samples ranged from 65.76% to 107.34% with a mean of 101.51% and 76.38% in high (7500 ng/mL) and low (30 ng/mL) QC concentrations, respectively. Recoveries of 2’-deoxyuridine in plasma samples ranged from 34.92% to 106.87% with a mean of 101.69% and 42.43% in high and low QC concentrations, respectively. Thymidine recoveries in urine samples ranged from 97.08% to 106.78% with mean of 101.28% and 103.34% in high (37.5 µg/mL) and low (3 µg/mL) QC concentrations, respectively. Recoveries of 2’-deoxyuridine in urine samples ranged from 101.98% to 112.98% with mean of 106.87% and 108.15% in high and low QC concentrations, respectively. No acceptance criteria were set for assay recoveries for two analytes. Lower recoveries were observed for both analytes at low concentration in plasma, however, the sensitivity of the MS detection helped to overcome the analyte loss and obtain necessary LLOQ values.

One of the limitations of using protein precipitation during sample preparation is the inefficient removal of matrix components which may co-elute and interfere with the analyte ionization process, by either suppressing or enhancing ion production. Table 2 shows the matrix effects and IS normalized matrix factors for thymidine and 2’-deoxyuridine in plasma, serum and urine tested at two QC concentrations for each assay. Signal suppression was observed for both thymidine and 2’-deoxyuridine in all plasma and serum matrices tested. The matrix effects or matrix factor according EMA guideline and IS normalized matrix factors lot-to-lot variability remained within the acceptance criteria for thymidine at both QC concentrations. For 2’-deoxyuridine, the lot-to lot variability remained within the set validation criteria for matrix effect at both QC concentrations and for IS normalized matrix factor at the higher QC concentration. The mean IS normalized MF% at the low QC concentration was 127.72% ranging, from 110.82% to 152.52%, with plasma with K_3_EDTA showing the largest matrix factor. Moreover, the variability of hyperlipaemic plasma was observed at 152.21%. Although hyperlipidaemia has been reported in some patients with MNGIE, the assay interference of lipids at this low concentration of 2’-deoxyuridine (equivalent to 0.12 µmol/L) will have no impact on the pharmacodynamic evaluation of EE-TP as the set criteria for metabolic correction in the clinical trial is a plasma 2’-deoxyuridine concentration 25 times higher than this, at <5 µmol/L (criteria for thymidine is a concentration <3 µmol/L) [14,21].

Signal suppression was observed for both thymidine and 2’-deoxyuridine for all six urines tested. The matrix effects and IS normalized matrix factors lot-to-lot variability remained within the acceptance criteria for both thymidine and 2’-deoxyuridine at both QC concentrations. 

### 3.5. Stability

Thymidine and 2’-deoxyuridine stability in plasma were evaluated after short term storage (ambient and refrigeration temperatures for 24 h), after three freeze-thaw cycles and after maintaining extracted samples on the autosampler for up to 72 h, see Table 3. The results are well within the acceptable limits of accuracy (within ± 15% of the actual concentration) and precision (CV ≤ ± 15%), demonstrating that the analytes are stable in plasma under conditions expected to be experienced using this analytical method.

No analytes were detected in urine after storage at ambient temperature for 24 h and at 4 °C for 14 days (Table 4). Microbial contamination is likely to be responsible for analyte degradation and this is therefore not an unexpected finding. Other studies report the immediate storage of collected urine at −20 °C to avoid analyte degradation [4]. To improve analyte stability, the addition of 5% PCA (v/v) to the urine samples was investigated. Although the addition of 5% PCA (v/v) was shown to improve the stability of thymidine and 2’-deoxyuridine, the stability results for thymidine at the high QC concentration were just outside the acceptable validation criteria. It is therefore recommended that urine samples are collected into containers spiked with preservative, are not stored at +4 °C or ambient temperature for prolonged periods of time and are frozen on receipt. The stability data for thymidine and 2’-deoxyuridine in urine samples stabilised with PCA and subjected to freeze-thaw cycles, and data from processed samples stored in the autosampler were within the set validation criteria for accuracy and precision.

Thymidine and 2’-deoxyuridine stock solution stability was evaluated by comparing analyte peak areas, in old and new stock solution. Over the storage period of 2.5 months the mean peak area of thymidine and 2’-deoxyuridine in old stock solution was 103.77% and 101.95%, respectively, of the peak area of newly prepared thymidine and 2’-deoxyuridine stock solution.

### 3.6. Carry-Over

No carry-over was observed for analytes and ISs in the plasma assay. In the urine assay, no carry-over was observed for analytes and 2’-deoxyuridine-1’,2’,3’,4’,5’-^13^C_5_. Carry-over for thymidine-1’,2’,3’,4’,5’-^13^C_5_ was below 2%, and therefore remained within the acceptable level of being less than 5% of the IS peak area. 

### 3.7. Clinical Application

To demonstrate the applicability of the plasma assay, thymidine and 2’-deoxyuridine concentrations were determined in the plasma of three patients with MNGIE, prior to therapy, and then during treatment with EE-TP. The plasma of two patients where a diagnosis of MNGIE was suspected were also analyzed. LC-MS/MS chromatograms of plasma from one patient pre-treatment and during treatment with EE-TP (day 117) are depicted in Figure 4 and show that measured thymidine and 2’-deoxyuridine concentrations were within the validated range. Table 5 shows the diagnosis concentrations of thymidine and 2’-deoxyuridine in five patients and the *TYMP* mutations which confirmed the diagnosis of MNGIE. 

The plasma concentrations of the analytes after treatment with EE-TP are shown for patients 1 to 3 and demonstrate that EE-TP is able to reduce or normalize the circulating concentrations of thymidine and 2’-deoxyuridine. These metabolite reductions coincided with clinical improvements. Patient 2 reported having a greater appetite during treatment through eating four meals per day. The patient also experienced tingling sensations in his feet, where previous to the start of therapy he had no sensation. During a clinical assessment by his neurologist specialist, improvements in the patient’s swallowing and dysgeusia were noted and his tongue looked less glossitic. Changes recorded for Patients 1 and 3 included decreased nausea and vomiting, increased walking distance, increases in the physical and mental components of the SF36 health and well-being survey, and improved sensory ataxia, balance and gait and fine finger movements [4,13].

A flexible dosing will be employed in the clinical trial of EE-TP with the aim of achieving metabolic correction, and thus following dosing and prior to the administration of subsequent doses, a pharmacodynamic evaluation will be necessary. For this reason, samples will not be subjected to long-term storage prior to analysis. However, to confirm the stability of analytes during the clinical trial sample storage period we are conducting a partial long-term stability validation. We have no reason to believe that there will be a deterioration in metabolite concentrations under the intended storage conditions and for the duration of the clinical trial, as plasma samples for patients 1 and 3 were analyzed after a storage period of over three years at −80 °C and provided measurements that were equivalent to those that had been measured by a UPLC we previously employed [4,13].

## 4. Conclusions

The LC–MS/MS methods presented in this paper were developed and validated for the simultaneous detection and quantification of thymidine and 2’-deoxyuridine in plasma and urine samples according to the guidelines of the EMA, with the aim of facilitating the pharmacodynamic evaluation of EE-TP, as part of a clinical trial for the treatment of patients with MNGIE. The plasma assay is the first validated LC–MS/MS assay reported which enables the simultaneous quantitation of thymidine and 2’-deoxyuridine in plasma with a lower limit of detection within the range of healthy individuals, and covers the metabolite concentration range expected from patients with late onset disease. The assay for the analysis of urine thymidine and 2’-deoxyuridine is the first reported validated method. Both methods met the acceptance criteria for selectivity with manageable matrix effects and good linearity, accuracy, precision, carry-over and recovery over the calibration range of 10 ng/mL to 10,000 ng/mL (0.04 to 41.28 µmol/L for thymidine and 0.04 to 43.82 µmol/L for 2’-deoxyuridine) in plasma, and 1 µg/mL to 50 µg/mL (0.004 to 0.21 mmol/L for thymidine and 0.004 to 0.22 mmol/L for 2’-deoxyuridine) in urine. The urine assay stability results support the use of preservative during collection and a nominal storage period at ambient and refrigeration temperatures. The methods use minimal plasma and urine volumes of 100 µL and 50 µL, respectively and this will particularly relevant to the paediatric patient cohorts where large plasma volumes would be unethical and impractical to obtain. Samples were prepared by protein precipitation and had short retention times of 2.0 and 1.2 min for the detection of thymidine and 2’-deoxyuridine, respectively, therefore providing easy to perform and rapid methods of analysis. The validated analytical method for plasma was successfully applied to the diagnosis of two patients with MNGIE (in synergy with mutation analyses) and to the pharmacodynamic assessment of three patients treated with EE-TP on a compassionate basis. Due to the logistics of collecting 24-h urine samples, the utility of the validated urine assay was not applied the measurement of urine metabolites in these patients. These assays will be used to support a clinical trial in patients with MNGIE through the longitudinal pharmacodynamic evaluation of EE-TP, with the results providing essential information for dose level selection and establishing therapeutic windows for treatment. 

## Figures and Tables

**Figure 1 jcm-09-00788-f001:**
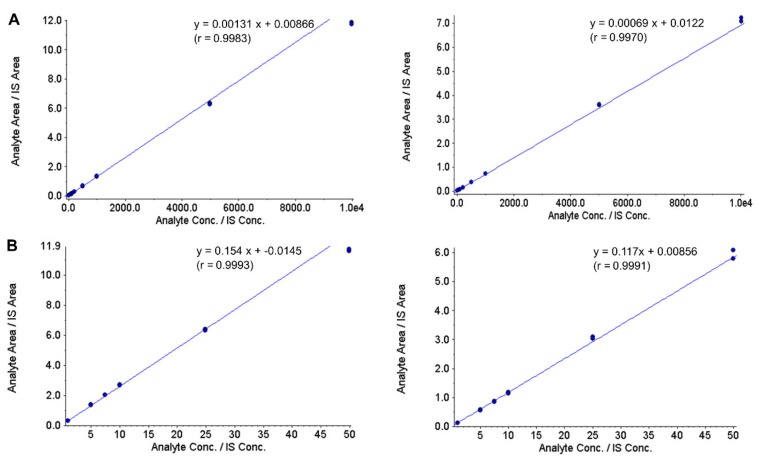
Representative calibration curves of thymidine (left panel) and 2’-deoxyuridine (right panel) in plasma (**A**) and urine (**B**) quantified by LC-MS/MS. Calibration standards ranged between 10 and 10,000 ng/mL in plasma and 1 and 50 µg/mL in urine.

**Figure 2 jcm-09-00788-f002:**
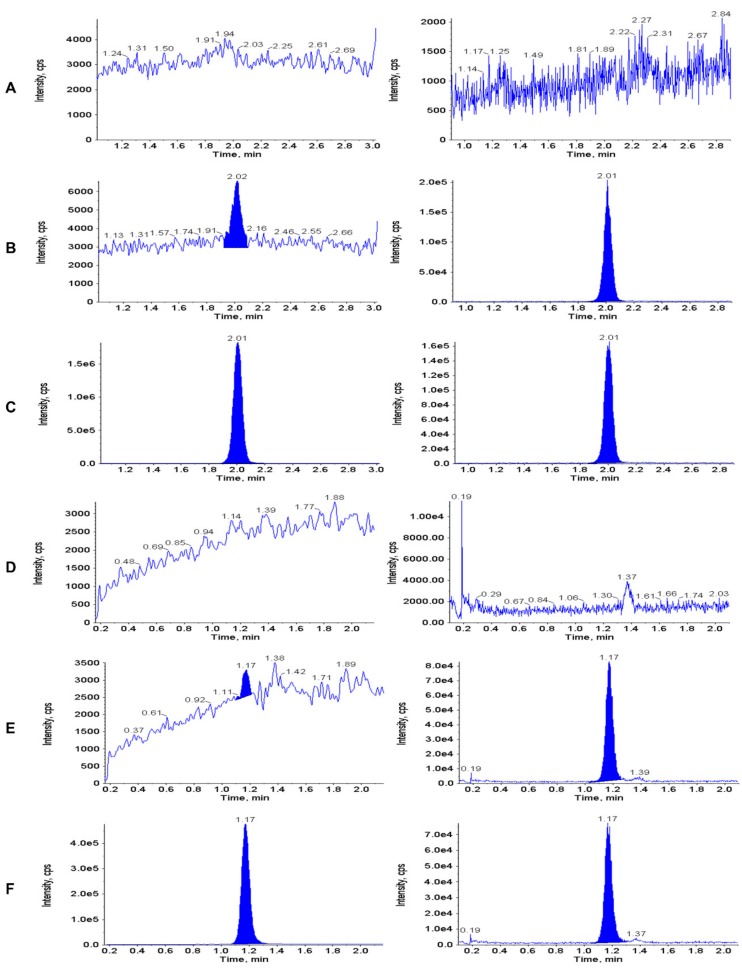
Representative chromatograms of extracted double blank plasma (**A** and **D**—0 ng/mL i.e. no analyte or IS), and plasma samples spiked with thymidine and 2’-deoxyuridine at the LLOQ (**B** and **E**—10 ng/mL) and upper limit of qualification (**C** and **F**—10,000 ng/mL). Thymidine: panels **A**, **B** and **C**; 2’-deoxyuridine: panels **D**, **E** and **F**).

**Figure 3 jcm-09-00788-f003:**
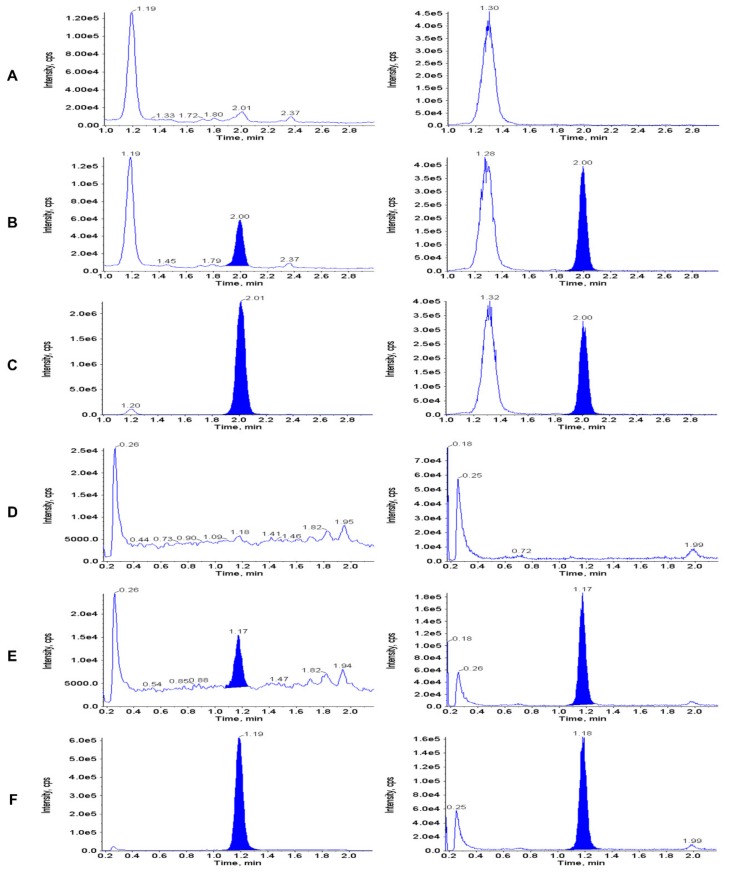
Representative chromatograms of extracted double blank urine (**A** and **D**—0 ng/mL i.e. no analyte or IS), and urine samples spiked with thymidine and 2’-deoxyuridine at the LLOQ (**B** and **E**: 1 µg/mL) and upper limit of qualification (**C** and **F**: 50 µg/mL). Thymidine: panels **A**, **B** and **C**; 2’-deoxyuridine: panels **D**, **E** and **F**.

**Figure 4 jcm-09-00788-f004:**
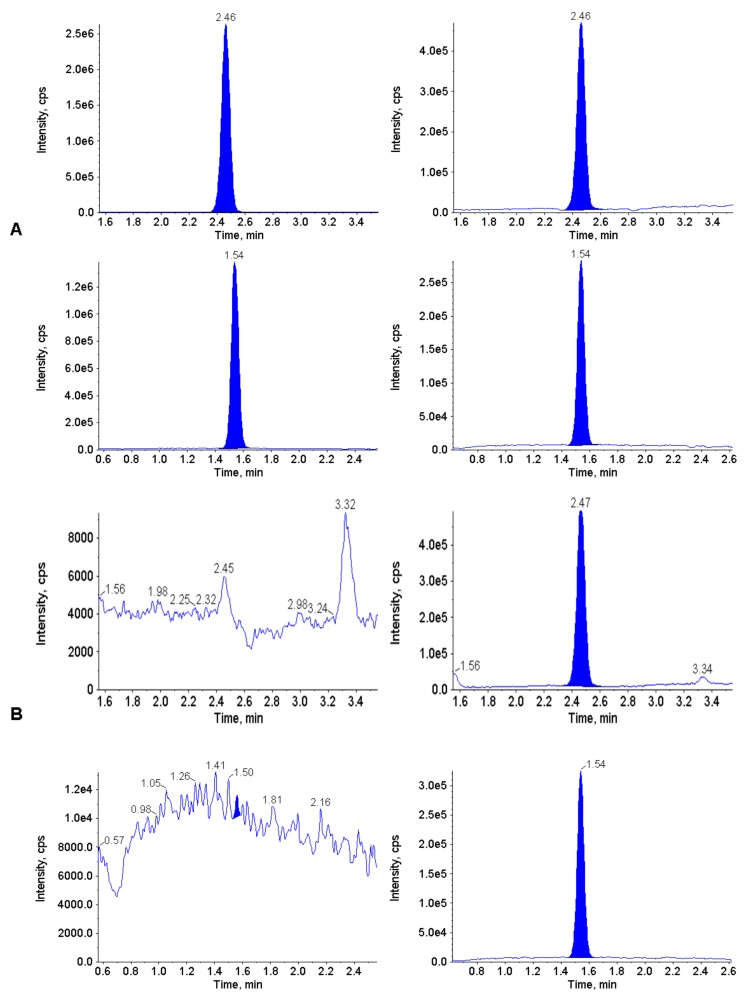
LC-MS/MS chromatograms of extracted plasma from patient 1 pre (**A**: upper panel thymidine; lower panel 2’deoxyuridine) and during treatment with EE-TP (**B**: upper panel thymidine; lower panel 2’deoxyuridine).

**Table 1 jcm-09-00788-t001:** Intra-day and inter-day accuracy and precision for urine and plasma assays. Each QC level was measured in six replicates on three different days.

Matrix and Analyte Concentration	Inter-Assay (*n* = 18) ^1^				Intra-assay (*n* = 6)		
	Spiked	Calculated Mean ± SD	Accuracy (%)	Precision (CV%)	Calculated Mean ± SD	Accuracy (%)	Precision (CV%)
Plasma Thymidine (ng/mL)	7515.0	7051.1 ± 197.1	93.83	2.80	7051.05 ± 97.32	93.83	1.38
	501.0	508.4 ± 10.6	101.47	2.08	508.39 ± 5.81	101.47	1.14
	30.06	31.5 ± 2.1	104.85	6.74	31.52 ± 1.63	104.85	5.18
	10.02	9.6 ± 1.2	95.64	12.17	9.59 ± 0.63	95.69	6.60
Plasma Deoxyuridine (ng/mL)	7522.5	7121.45 ± 284.27	94.67	3.99	7121.45 ± 254.70	94.67	3.58
	501.5	481.84 ± 24.97	96.08	5.18	481.84 ± 14.43	96.08	3.00
	30.09	29.71 ± 2.57	98.72	8.65	29.63 ± 1.38	98.49	4.66
	10.03	10.42 ± 1.23	103.91	11.83	10.39 ± 0.46	103.63	4.40
Urine Thymidine (µg/mL)	37.58	36.13 ± 1.98	96.14	5.48	36.13 ± 2.24	96.14	6.19
	15.03	15.36 ± 0.72	102.43	4.70	15.36 ± 0.78	102.43	5.08
	3.01	3.22 ± 0.12	106.99	3.70	3.23 ± 0.14	107.30	4.36
	1.00	1.03 ± 0.10	103.09	9.50	1.05 ± 0.12	104.80	11.51
Urine Deoxyuridine (µg/mL)	37.61	39.02 ± 1.50	103.76	3.84	38.57 ± 1.50	102.55	3.89
	15.05	15.56 ± 0.80	103.40	5.15	15.41 ± 0.91	102.36	5.88
	3.01	3.13 ± 0.15	104.03	4.89	3.14 ± 0.16	104.37	5.04
	1.00	1.06 ± 0.08	106.20	7.09	1.07 ± 0.09	107.45	8.18

^1^ Inter-assay (*n* = 18): each QC level was measured in six replicates on three different days.

**Table 2 jcm-09-00788-t002:** Matrix effects (ME%) or matrix factor (MF%) and IS normalized matrix factors for thymidine and 2’-deoxyuridine in plasma, serum and urine tested at two QC concentrations.

Matrix	Thymidine				2’-deoxyuridine			
	Average ME% or MF%		Average IS Normalized MF%		Average ME% or MF%		Average IS normalized MF%	
	High QC (*n* = 6)	Low QC (*n* = 6)	High QC (*n* = 6)	Low QC (*n* = 6)	High QC (*n* = 6)	Low QC (*n* = 6)	High QC (*n* = 6)	Low QC (*n* = 6)
Plasma with K_3_EDTA	76.65	80.13	97.58	114.74	51.59	66.62	106.92	152.52
Dialyzed plasma with K_2_EDTA	70.74	84.74	100.40	101.79	48.09	46.38	108.63	110.82
Plasma with Lithium Heparin (1)	68.95	69.13	100.08	102.94	47.33	55.73	111.28	131.89
Plasma with Lithium Heparin (2)	68.54	71.59	98.96	107.55	46.52	46.98	107.23	111.81
Serum (1)	69.74	65.91	99.69	97.51	47.27	55.89	106.02	127.48
Serum (2)	71.12	70.19	98.70	103.14	49.12	56.27	112.47	131.78
Mean	70.96	73.62	99.24	104.61	48.32	54.65	108.76	127.72
Hyperlipidaemic plasma	67.87	67.89	101.21	102.77	48.09	65.65	108.68	152.21
Haemolysed plasma with 2.5% blood	69.84	67.38	102.00	103.11	48.48	43.91	110.11	103.48
Haemolysed plasma with 5% blood	68.10	68.81	99.74	105.07	47.04	49.95	109.00	120.11
Haemolysed plasma with 7.5% blood	68.76	68.48	101.46	105.42	46.43	43.33	107.99	105.11
Haemolysed plasma with 10% blood	70.49	67.06	101.62	101.89	47.51	47.85	107.76	112.61
Mean	69.01	67.92	101.65	10365	47.51	50.14	108.71	118.70
Urine (1)	78.07	65.53	103.29	101.48	48.27	38.03	103.14	98.55
Urine (2)	74.64	64.70	103.50	101.91	44.79	35.68	104.01	100.88
Urine (3)	71.82	59.36	104.31	104.08	32.24	22.16	98.82	89.17
Urine (4)	82.28	66.41	102.10	97.56	55.72	43.19	106.17	102.12
Urine (5)	76.43	64.13	104.98	102.76	47.17	37.76	105.38	102.99
Urine (6)	77.15	62.76	105.63	100.43	47.75	38.22	103.86	100.77
Mean	76.73	63.82	103.97	101.37	45.99	35.84	103.84	99.08

Thymidine: High QC = 7500 ng/mL, Low QC = 30 ng/mL; 2’deoxyuridine: High QC = 37.5 µg/mL, Low QC = 3 µg/mL.

**Table 3 jcm-09-00788-t003:** Stability testing of thymidine and 2’-deoxyuridine in plasma.

Stability Test	Thymidine				2’-deoxyuridine			
	Concentration (ng/mL)		CV%	Accuracy (%)	Concentration (ng/mL)		CV%	Accuracy (%)
	Spiked	Mean ± SD			Spiked	Mean ± SD		
Time 0	30.06	30.79 ± 1.70	5.54	102.42	30.09	30.77 ± 2.91	9.44	102.27
	7515.0	6942.69 ± 171.60	2.47	92.38	7522.5	7358.44 ± 83.49	1.13	97.82
24 h RT	30.06	30.21 ± 2.02	6.69	100.50	30.09	30.12 ± 2.80	9.30	100.11
	7515.0	7017.00 ± 148.35	2.11	93.37	7522.5	7039.48 ± 149.61	2.13	93.58
24 h 4 °C	30.06	31.29 ± 1.21	3.88	104.08	30.09	30.00 ± 3.92	13.08	99.71
	7515.0	7145.74 ± 206.99	2.90	95.09	7522.5	6918.37 ± 96.97	1.40	91.97
Cycle 1	30.06	30.57 ± 2.31	7.57	101.71	30.09	30.19 ± 2.54	8.42	100.32
	7515.0	7258.70 ± 141.88	1.95	96.59	7522.5	6852.73 ± 162.53	2.37	91.10
Cycle 2	30.06	31.64 ± 2.36	7.45	105.24	30.09	30.45 ± 2.65	8.69	101.19
	7515.0	7077.90 ± 180.94	2.56	94.18	7522.5	7057.16 ± 243.80	3.45	93.81
Cycle 3	30.06	32.98 ± 0.97	2.94	109.70	30.09	31.79 ± 2.90	9.12	105.66
	7515.0	7018.79 ± 111.02	1.58	93.40	7522.5	7228.30 ± 126.25	1.75	96.09
Time 0	10.02	10.26 ± 1.40	13.62	102.56	10.03	9.88 ± 1.36	13.78	98.83
	30.06	30.79 ± 1.70	5.54	102.42	30.09	30.77 ± 2.91	9.44	102.27
	501.0	501.69 ± 4.30	0.86	100.14	501.5	495.66 ± 29.25	5.90	98.83
	7515.0	6942.69 ± 171.60	2.47	92.38	7522.5	7358.44 ± 83.49	1.13	97.82
24 h 4 °C	10.02	9.84 ± 0.91	9.20	98.37	10.03	10.34 ± 1.02	9.89	103.43
	30.06	32.04 ± 1.77	5.51	106.59	30.09	30.08 ± 2.47	8.21	99.97
	501.0	493.44 ± 13.29	2.69	98.49	501.5	469.55 ± 4.81	1.03	93.63
	7515.0	6805.05 ± 43.87	0.65	90.55	7522.5	6841.37 ± 136.66	2.00	90.95
48 h 4 °C	10.02	10.37 ± 1.11	10.74	103.73	10.03	10.55 ± 0.60	5.65	105.52
	30.06	30.75 ± 0.79	2.57	102.28	30.09	31.87 ± 1.86	5.83	105.93
	501.0	503.37 ± 8.29	1.65	100.47	501.5	484.65 ± 9.68	2.00	96.64
	7515.0	6902.64 ± 203.18	2.94	91.85	7522.5	7136.11 ± 166.93	2.34	94.86
72 h 4 °C	10.02	10.30 ± 0.44	4.30	102.97	10.03	9.89 ± 1.07	10.83	98.93
	30.06	30.62 ± 2.73	8.93	101.85	30.09	30.58 ± 2.00	6.55	101.64
	501.0	500.08 ± 11.53	2.31	99.82	501.5	486.86 ± 15.87	3.26	97.08
	7515.0	6879.71 ± 244.09	3.55	91.55	7522.5	7102.80 ± 251.75	3.54	94.42

RT = room temperature.

**Table 4 jcm-09-00788-t004:** Stability testing of thymidine and 2’-deoxyuridine in urine.

Stability Test	Thymidine				2’-Deoxyuridine			
	Concentration (µg/mL)		CV%	Accuracy (%)	Concentration (µg/mL)		CV%	Accuracy (%)
	Spiked	Mean ± SD			Spiked	Mean ± SD		
**Short term**								
Time 0	3.01	3.19 ± 0.03	0.89	105.99	3.01	3.01 ± 0.07	2.40	103.07
	37.58	34.96 ± 0.51	1.47	93.03	37.61	38.97 ± 1.57	4.02	103.62
24 h RT + PCA	3.01	2.92 ± 0.05	1.66	97.05	3.01	3.10 ± 0.10	3.19	103.08
	37.58	31.32 ± 0.54	1.73	83.34	37.61	36.98 ± 0.95	2.57	98.33
24 h RT	3.01	ND			3.01	ND		
	37.58	ND			37.61	ND		
14 days 4 °C + PCA	3.01	3.03 ± 0.17	5.71	100.75	3.01	2.69 ± 0.16	5.96	89.26
	37.58	30.15 ± 0.63	2.07	80.24	37.61	33.42 ± 1.13	3.38	88.87
14 days 4 °C	3.01	ND			3.01	ND		
	37.58	ND			37.61	ND		
**Freeze-thaw**								
Cycle 1 + PCA	3.01	3.18 ± 0.09	2.85	105.68	3.01	3.07 ± 0.07	2.26	102.08
	37.58	34.74 ± 0.44	1.27	92.44	37.61	38.43 ± 0.82	2.15	102.19
Cycle 2 + PCA	3.01	3.24 ± 0.10	3.24	107.57	3.01	3.23 ± 0.13	4.02	107.31
	37.58	35.76 ± 0.63	1.77	95.17	37.61	39.56 ± 0.67	1.69	105.17
Cycle 3 + PCA	3.01	3.24 ± 0.05	1.43	107.70	3.01	3.01 ± 0.10	3.43	100.08
	37.58	35.66 ± 0.71	1.99	94.88	37.61	37.82 ± 0.82	2.16	100.55
Cycle 1	3.01	2.82 ± 0.06	2.10	93.82	3.01	2.46 ± 0.05	1.87	81.84
	37.58	33.56 ± 0.53	1.58	89.29	37.61	35.09 ± 0.50	1.42	93.31
Cycle 2	3.01	2.80 ± 0.08	2.72	92.90	3.01	2.43 ± 0.05	1.98	80.76
	37.58	33.39 ± 0.80	2.40	88.85	37.61	34.73 ± 0.72	2.07	92.34
Cycle 3	3.01	2.83 ± 0.06	2.02	93.89	3.01	2.37 ± 0.06	2.73	78.73
	37.58	33.91 ± 0.81	2.38	90.23	37.61	33.35 ± 0.66	1.97	88.67
**Autosampler**								
Time 0	1.00	1.19 ± 0.01	0.84	118.50	1.00	1.18 ± 0.02	1.77	117.50
	3.01	3.39 ± 0.03	0.85	112.49	3.01	3.32 ± 0.06	1.89	110.17
	15.03	16.23 ± 0.24	1.45	108.00	15.05	116.44 ± 0.36	2.20	109.20
	37.58	38.71 ± 0.81	2.08	103.00	37.61	40.30 ± 0.71	1.75	107.14
24 h 4 °C	1.00	1.14 ± 0.05	3.97	113.50	1.00	1.13 ± 0.07	5.84	113.43
	3.01	3.28 ± 0.02	0.61	108.97	3.01	3.26 ± 0.15	4.52	108.45
	15.03	16.04 ± 0.30	1.90	106.72	15.05	15.92 ± 0.33	2.10	105.75
	37.58	37.83 ± 0.77	2.04	100.67	37.61	38.11 ± 0.50	1.32	101.33
48 h 4 °C	1.00	1.14 ± 0.04	3.52	114.30	1.00	1.09 ± 0.10	8.72	109.38
	3.01	3.32 ± 0.08	2.37	110.43	3.01	3.17 ± 0.12	3.70	105.26
	15.03	16.60 ± 0.34	2.06	110.47	15.05	15.21 ± 0.32	2.14	101.06
	37.58	38.38 ± 1.23	3.21	102.12	37.61	38.57 ± 1.04	2.69	102.56
72 h 4 °C	1.00	1.14 ± 0.02	1.93	114.12	1.00	1.18 ± 0.01	1.19	118.10
	3.01	3.32 ± 0.04	1.21	110.33	3.01	3.32 ± 0.02	0.64	110.26
	15.03	16.27 ± 0.18	1.09	108.24	15.05	15.94 ± 0.31	1.98	105.88
	37.58	37.63 ± 1.13	2.99	100.13	37.61	39.38 ± 0.57	1.44	104.70

ND = not detected; RT = room temperature; PCA = perchloric acid.

**Table 5 jcm-09-00788-t005:** Plasma thymidine and 2’deoxyuridine concentrations in patients with MNGIE prior to therapy and during therapy with EE-TP (patients 1 to 3) and in patients tested for MNGIE (patients 4 and 5).

Patient ID	*TYMP* Mutation	Diagnosis Concentration, ng/mL (µmol/L)		EE-TP dose (µmol/min/10^10^ Cells)	Treatment Day	In-Treatment Concentration, ng/mL (µmol/L)	
		Thymidine	2’deoxyuridine			Thymidine	2’deoxyuridine
1	Heterozygous c.866A>C c.1231_1243 del	5536.5 (22.9)	6161.3 (27.0)	18.2	117	ND	ND
2	Homozygous c.1088 delG	3029.5 (12.5)	3638.1 (15.9)	26.5	33	40.7 (0.17)	9.9 (0.4)
3	g.4009_4010insG g.4101G>A	4031.5 (16.6)	4817.6 (21.1)	27.5	34	2068.5 (8.5)	2251.0 (9.9)
4	Homozygous c.1282G>C	2380.9 (9.8)	2921.8 (12.8)	NA	NA	NA	NA
5	Homozygous c.392C>T	2892.0 (11.9)	2558.4 (11.2)	NA	NA	NA	NA

ND = Not detected; NA = Not applicable; EE-TP = erythrocyte encapsulated thymidine phosphorylase.

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
