# Peer review of "Quantification of Plasma and Urine Thymidine and 2’-Deoxyuridine by LC-MS/MS for the Pharmacodynamic Evaluation of Erythrocyte Encapsulated Thymidine Phosphorylase in Patients with Mitochondrial Neurogastrointestinal Encephalomyopathy"

_jcm, 2020, doi:10.3390/jcm9030788_

Round 1

Reviewer 1 Report

The Authors detailed a bioanalytical assay to determine the urine and plasma levels of thymidine and 2’-deoxyuridine using a liquid chromatography- tandem mass spectrometry (LC–MS/MS). This method was developed according to regulatory guidelines to provide accurate assessment of erythrocyte encapsulated thymidine phosphorylase (EE-TP), a novel temporary therapeutic measure in patients with mitochondrial neurogastrointestinal encephalomyopathy (MNGIE).

The method presented in this paper is accurately described and explored. The results obtained are reported in detail and they appear reproducible. An equivalent method had already been published in the literature, however, this second work showed an increased detection range, capable of distinguishing very low concentrations typical of controls and is therefore more useful in detecting late onset patients with MNGIE.

It is a good article and certainly indications to the analysis of circulating nucleotides and clinical monitoring of patients with MNGIE. Also, LC- MS/MS represents the first validated method for the quantification of thymidine and 2’-deoxyuridine in biological fluids other than plasma, such as urine (possibly easier to be collected).

I have only few minor comments:

1) Line 397- 402 Authors indicate clinical improvements after erythrocyte encapsulated TP (EE-TP ) therapy. “The plasma concentrations of the analytes after treatment with EE-TP are shown for patients 1 to 3 and demonstrate that EE-TP is able to reduce or normalise the circulating concentrations of thymidine and 2’-deoxyuridine. These metabolite reductions coincided with clinical improvements. Patient 2 showed an increase in body weight, from 40.6 kg to 40.8 kg between pre-therapy and the fourth treatment cycle administered and reported having a greater appetite through eating 4 meals per day.” I agree that the treatment ameliorate the biochemical condition since EE-TP normalise the circulating concentrations of thymidine and 2’-deoxyuridine. However, the body weight increase (0.2 Kg) of patient #2 was minimal at best or virtually absent. I suggest not considering this as a clinical improvement and removing it from the text. This info would, in my humble opinion, reduce the impact of this excellent work.

2) Line 405-406 “Clinical improvements in patients 1 and 3 have been reported elsewhere [4,13]”. As the paragraph focuses on clinical improvements of the patients treated, I suggest that a very brief summary of these improvements should be added (and not only referenced); this would avoid the readers to look for these reported data.

3) The captions in the various Figures cannot be seen at all ! Either delete or increase the font.

Author Response

The authors thank the reviewer for his/her feedback and for giving up their time for reviewing our manuscript.

Please find our responses to the feedback provided:

1) Line 397- 402 Authors indicate clinical improvements after erythrocyte encapsulated TP (EE-TP) therapy. “The plasma concentrations of the analytes after treatment with EE-TP are shown for patients 1 to 3 and demonstrate that EE-TP is able to reduce or normalise the circulating concentrations of thymidine and 2’-deoxyuridine. These metabolite reductions coincided with clinical improvements. Patient 2 showed an increase in body weight, from 40.6 kg to 40.8 kg between pre-therapy and the fourth treatment cycle administered and reported having a greater appetite through eating 4 meals per day.” I agree that the treatment ameliorate the biochemical condition since EE-TP normalise the circulating concentrations of thymidine and 2’-deoxyuridine. However, the body weight increase (0.2 Kg) of patient #2 was minimal at best or virtually absent. I suggest not considering this as a clinical improvement and removing it from the text. This info would, in my humble opinion, reduce the impact of this excellent work.

We have removed the reference to weight increase, please see lines 404-406.

2) Line 405-406 “Clinical improvements in patients 1 and 3 have been reported elsewhere [4,13]”. As the paragraph focuses on clinical improvements of the patients treated, I suggest that a very brief summary of these improvements should be added (and not only referenced); this would avoid the readers to look for these reported data.

We have included the following sentence (lines 409-413): “Changes recorded for Patients 1 and 3 included decreased nausea and vomiting, increased walking distance, increases in the physical and mental components of the SF36 health and well-being survey, and improved sensory ataxia, balance and gait and fine finger movements [4 ,13]

3) The captions in the various Figures cannot be seen at all ! Either delete or increase the font.

We have increased the font size for fig 1 and removed the Figure captions on Figures 2, 3 and 4.

Reviewer 2 Report

The authors reported a method for the determination of thymidine and 2’-deoxyuridine in plasma and urine. Liquid chromatography coupled with tandem mass spectrometry was used. The method was validated in accordance with EMA guidelines. The method was reported in detail and the validation procedure was extensively described.  

The method was applied to the diagnosis and treatment monitoring of mitochondrial neurogastrointestinal encephalomyopathy (MNGIE), a rare disorder caused by a defect of thymidine phosphorylase.  Currently, new treatments for MNGIE have been under investigation. In particular this method could be useful to investigate the efficacy of EE-TP in patients with MNGIE, in the hypothesis that treatment with EE-TP will arrest and reverse the progression of the clinical disease

Comments and recommendations:

  • Sample preparation line 142-145:Was sample preparation procedure used for patient’s plasma samples too? Protein precipitation by acidification cannot be considered a extraction of analytes (thymidine and 2’-deoxyuridine) from samples (QC , calibrators etc), therefore the term “extraction” is not appropriate.  
  • Conclusions line 425: the author reported ”The assay for analysis of urine thymidine and 2’-deoxyuridine is the first reported validated method.” However clinical validity of urine thymidine and 2’-deoxyuridine assay was not tested since it was not applied on patient’s urine sample. Authors should comment this point.
  • Line 130 unit (μg/mL) was reported 2 times
  • Line 164 TIS is the acronym for TurboIonSpray
  • Fig 1 A The highest points and their linear interpolation were not clearly displayed
  • Line 301 It is not clear in which way the analyte selectivities (0%) were calculated
  • Table 1 Are analyte concentrations in urine expressed in ng/mL?
  • Recovery Thymidine and 2’-deoxyuridine mean recoveries obtained in low QC is quite low, near 50% of analyte was lost during protein precipitation. Authors should comment this result.

Author Response

The authors thank the reviewer for his/her feedback and for giving up their time for reviewing our manuscript.

Please find our responses to the feedback provided:

  1. Sample preparation line 142-145: Was sample preparation procedure used for patient’s plasma samples too? Protein precipitation by acidification cannot be considered a extraction of analytes (thymidine and 2’-deoxyuridine) from samples (QC , calibrators etc), therefore the term “extraction” is not appropriate.

We have removed the term extraction and the sentences now read as follows (see Lines 144-151):

Plasma samples were prepared by adding 100 μL of working IS solution to 100 µL of each plasma Calibrator standard (except double blank, i.e. plasma sample containing no analyte or IS), QC standard and patient sample, followed by the addition of 100 μL of 7% PCA in deionised water.  Urine samples were prepared by adding 50 μL of working IS solution to 50 µL of each Calibrator (except double blank, i.e. urine sample containing no analyte or IS) and QC standard, followed by the addition of 125 μL of 7% PCA in deionised water.

  1. Conclusions line 425: the author reported ”The assay for analysis of urine thymidine and 2’-deoxyuridine is the first reported validated method.” However clinical validity of urine thymidine and 2’-deoxyuridine assay was not tested since it was not applied on patient’s urine sample. Authors should comment this point.

We have included the following sentence (lines ): “Due to the logistics of collecting 24-hour urine samples, the utility of the validated urine assay was not applied the measurement of urine metabolites in these patients.”

  1. Line 130 unit (μg/mL) was reported 2 times

We have removed the duplication

  1. Line 164 TIS is the acronym for TurboIonSpray

We have removed the acronym and inserted the full name.

  1. Fig 1 A The highest points and their linear interpolation were not clearly displayed

We have improved Figure 1 as requested.

  1. Line 301 It is not clear in which way the analyte selectivities (0%) were calculated

The calculation of assay selectivity is described in lines 191 to 198. We have improved the manuscript to clarify how the selectivity is calculated as follows:

For the evaluation of selectivity, six blank human urine samples, nine blank plasma samples (dialysed K2EDTA plasma and plasma containing K3EDTA, lithium heparin, cholesterol, 2.5%, 5%, 7.5% and 10% lysed whole blood) and two serum samples from different donors were individually analysed and evaluated for interference using the proposed extraction procedure and chromatographic/mass spectrometric conditions, and compared with the responses obtained with calibrators containing the analytes at concentrations near to the LLOQ. The absence of interfering components will be accepted where the response from interfering compounds is less than 20 % of the LLOQ for the analyte and less than 5% for the IS. (lines 193-200)

  1. Table 1 Are analyte concentrations in urine expressed in ng/mL?

Yes, thank you for spotting this.  We have corrected this, please see Table 1.

  1. Recovery Thymidine and 2’-deoxyuridine mean recoveries obtained in low QC is quite low, near 50% of analyte was lost during protein precipitation. Authors should comment this result.

Thank you for the suggestion. We have added the following (lines 347-349):

No acceptance criteria were set for assay recoveries for two analytes. Lower recoveries were observed for both analytes at low concentration in plasma, however, the sensitivity of the MS detection helped to overcome the analyte loss and obtain necessary LLOQ values.

Reviewer 3 Report

The authors describe a LC-MS/MS method for the quantification of plasma and urine thymidine and 2'deoxyuridine applicable to patients with MNGIE. They outline among the strenghts the achievement of a lower limit of detection and method validation procedure according to EMA guidelines.

Major observations:

  • Introduction
  • page 2, line 73: there is not a "normal range" for plasma thymidine (dThd) and deoxyuridine(dUrd). It is more appropriate to say that the two analytes are undetectable in unaffected subjects below the commonly reported limit of 0.05 umol/L. 
  • Method validation,
  • a) page 4, line 173, ref. 15: please refer to the original document of EMA guidelines; 
  • b) page 5, line 201: for calculations of interassay and intrassay accuracies spiked concentrations for each QC are mentioned, while in “Plasma assay paragraph” (page 3, lines 105-106) standard concentrations in dialysed human plasma QCs are reported. Please explain these discrepancies;
  • c) page 5, lines 209 and following: the description of matrix effect experiments is quite confused and should be revised. In particular, what was specifically done to calculate the matrix effect as described by Matuszewski? On the other hand, along with EMA guidelines “Matrix effects should be investigated using at least 6 lots of blank matrix from individual donors”. This means that 6 different lots should be analyzed for each type of source (i.e. K3EDTA plasma, lithium heparin plasma, serum) and not six different plasma samples. Averaged matrix effect values based on single assessments of so different sources can be misleading. Even haemolysed and hyperlipidaemic plasma samples should be considered in addition to the normal matrix, and not included in the average calculations. Please revise also the calculation of IS normalised MF following the EMA guidelines:“The IS normalised MF should also be calculated by    dividing the MF of the analyte by the MF of the IS”;  
  • d) page 6, lines 248-249: long term stability of the analytes in plasma samples stored in the freezer was not checked and must be addressed. This information is important also to evaluate the accuracy of the results from patients’ plasma specimens that were “collected and stored at -80°C until analysis” (page 6, line 263).
  • Results and Discussion
  • a) page 6, line 265: reported retention times for dThd and dUrd in plasma and urine are 2 min and 1.2 min, respectively. Looking at plasma chromatograms a large intersamples variability in retention time for both analytes is evident: dThd, 1.95, 2.28 (Figure 2), 2.48 (Figure 4); dUrd, 1.15, 1.43 (Figure 2) and 1.54 (Figure 3). Did you examine the potential causes of this variability? Could it affect assay selectivity? This issue must be discussed in the text.
  • b) page 10, lines 329-332: recovery ranges show a large difference between low and high QCs for both analytes. In particular, recovery values for dUrd low QCs are about a third compared with high QCs. Although no acceptance criteria for recovery are mentioned in EMA guidelines, low recovery values can contribute to assay inaccuracy and imprecision. How did you explain and cope with this low recovery for low QCs? Did you establish any acceptance requirements for recovery values? Moreover, recovery is generally explored at low, medium and high concentration values within the calibration range.

Minor observations:

  • In Table 2, IS normalised matrix factor are detailed. It should be more appropriate to indicate these values as IS-MF% in place of MF%.
  • Figures 2-3-4: Chromatograms are reported in a very reduced size and numeric values of x and y axes are near unreadable. It is recommended to improve chromatograms reproduction.
  • Table 3, please specify 0 as Time 0.

Author Response

The authors thank the reviewer for his/her feedback and for giving up their time for reviewing our manuscript.

Please find our responses to the feedback provided:

Major observations:

Introduction

Page 2, line 73: there is not a "normal range" for plasma thymidine (dThd) and deoxyuridine(dUrd). It is more appropriate to say that the two analytes are undetectable in unaffected subjects below the commonly reported limit of 0.05 umol/L. 

We have amended this sentence to:

However, plasma concentrations of thymidine and 2’-deoxyuridine in unaffected individuals are undetectable, being below the commonly reported limit of 0.05 umol/L.

 Method validation

a)page 4, line 173, ref. 15: please refer to the original document of EMA guidelines;

 We have referred to the original document, please see reference 15.

b) page 5, line 201: for calculations of interassay and intrassay accuracies spiked concentrations for each QC are mentioned, while in “Plasma assay paragraph” (page 3, lines 105-106) standard concentrations in dialysed human plasma QCs are reported. Please explain these discrepancies;

Under Materials and Methods nominal concentrations have been stated. Accuracy and precision were calculated using actual spiked concentrations. Manuscript text has been changed as follows to improve the readability (page 3, lines 107-110):

Calibrator standards were prepared by appropriate dilution of the Calibrator sub-stock solutions with analyte-free dialysed human plasma to contain 10, 50, 100, 200, 500, 1,000, 5,000 and 10,000 ng/mL of thymidine and 2’-deoxyuridine (nominal concentrations). QC standards were similarly prepared by the appropriate dilution of the combined QC sub-stocks at concentrations 10 µg/mL and 0.1 µg/mL. QC standard concentrations in dialysed human plasma were 10, 30, 500 and 7,500 ng/mL of thymidine and 2’-deoxyuridine (nominal concentrations).

c) page 5, lines 209 and following: the description of matrix effect experiments is quite confused and should be revised. In particular, what was specifically done to calculate the matrix effect as described by Matuszewski? On the other hand, along with EMA guidelines “Matrix effects should be investigated using at least 6 lots of blank matrix from individual donors”. This means that 6 different lots should be analyzed for each type of source (i.e. K3EDTA plasma, lithium heparin plasma, serum) and not six different plasma samples. Averaged matrix effect values based on single assessments of so different sources can be misleading. Even haemolysed and hyperlipidaemic plasma samples should be considered in addition to the normal matrix, and not included in the average calculations. Please revise also the calculation of IS normalised MF following the EMA guidelines:“The IS normalised MF should also be calculated by    dividing the MF of the analyte by the MF of the IS”; 

Thank you for your comment. Yes, matrix effects were evaluated according Internal standard normalised matrix factor was evaluated according to EMA guidelines.

The manuscript has been amended as follows (page 5, lines 214- 216): The potential suppression/enhancement of thymidine and 2’-deoxyuridine ionization by matrix components during LC–MS/MS analysis was evaluated by measuring the matrix effects as described by Matuszewski et al [21] or as matrix factor described by EMA guidelines.

Table 2 has been changed to reflect the changes requested by the reviewer.

Manuscript text has been changed as follows (page 14, lines 359-362): The matrix effects or matrix factor according EMA guideline and IS normalised matrix factors lot-to-lot variability remained within the acceptance criteria for thymidine at both QC concentrations. For 2’-deoxyuridine, the lot-to lot variability remained within the set validation criteria for matrix effect at both QC concentrations and for IS normalised matrix factor at the higher QC concentration. The mean IS normalized MF% at the low QC concentration was 127.72% ranging, from 110.82% to 152.52%, with plasma with K3EDTA showing the largest matrix factor. Moreover, the variability of hyperlipaemic plasma was observed at 152.21%.

Thank you for pointing out the need for matrix effects evaluation in different matrices and in six individual sources for each matrix. This was not conducted. However, the method validations conducted were reviewed by the regulatory bodies and was deemed acceptable for this study, particularly as different counterions are considered to represent the same anticoagulant.

d) page 6, lines 248-249: long term stability of the analytes in plasma samples stored in the freezer was not checked and must be addressed. This information is important also to evaluate the accuracy of the results from patients’ plasma specimens that were “collected and stored at -80°C until analysis” (page 6, line 263).

Thank you for your comment. Long-term stability study was not completed during this validation, but is currently being conducted separately as partial validation to ensure the stability of analytes during the clinical trial sample storage period.  In the clinical trial, the pharmacodynamic evaluation of EE-TP will require the analysis of samples within a week of collection to enable dose level selection for the next treatment cycle and for defining therapeutic windows for treatment. The plasma samples analyzed from treated patients reported here had been stored for over three years at -80°C, and provided measurements that were equivalent to those that had been measured by a UPLC we previously employed  (see reference 4).

 Results and Discussion

a) page 6, line 265: reported retention times for dThd and dUrd in plasma and urine are 2 min and 1.2 min, respectively. Looking at plasma chromatograms a large intersamples variability in retention time for both analytes is evident: dThd, 1.95, 2.28 (Figure 2), 2.48 (Figure 4); dUrd, 1.15, 1.43 (Figure 2) and 1.54 (Figure 3). Did you examine the potential causes of this variability? Could it affect assay selectivity? This issue must be discussed in the text.

The change in the retention was investigated. The change came from the first batch of analysis when the column was not properly equilibrated and therefore the retention time shifted a little. The batch passed regardless of the change in the retention, and therefore it was accepted and not repeated. This also means that the changes in the retention time did not affect the analysis.

The change in the retention time was not observed later in the following batches. We have changed figures 2 and presented now the chromatograms from batch 2.

b) page 10, lines 329-332: recovery ranges show a large difference between low and high QCs for both analytes. In particular, recovery values for dUrd low QCs are about a third compared with high QCs. Although no acceptance criteria for recovery are mentioned in EMA guidelines, low recovery values can contribute to assay inaccuracy and imprecision. How did you explain and cope with this low recovery for low QCs? Did you establish any acceptance requirements for recovery values? Moreover, recovery is generally explored at low, medium and high concentration values within the calibration range.

Thank you for your comment. We have added the following to the manuscript: (lines 347-350):

No acceptance criteria were set for assay recoveries for two analytes. Lower recoveries were observed for both analytes at low concentration in plasma, however, the sensitivity of the MS detection helped to overcome the analyte loss and obtain necessary LLOQ values.

Recovery is not a requirement of the EMA, and so only two concentrations were explored.

Minor observations:

In Table 2, IS normalised matrix factor are detailed. It should be more appropriate to indicate these values as IS-MF% in place of MF%.

We have changed Table 2 as requested.

Figures 2-3-4: Chromatograms are reported in a very reduced size and numeric values of x and y axes are near unreadable. It is recommended to improve chromatograms reproduction.

All figures have been changed as requested.

Table 3, please specify 0 as Time 0.

Both Table 3 and 4 have been changed as requested.

Round 2

Reviewer 3 Report

The authors have answered satisfactorily to all observations.

As a further suggestion, a comment on available data (both in their possession, both published) on analytes' long term stability at -80°C should be added. 

Author Response

We thank the referee for accepting our responses and for his/her time in providing a second review.

We provide the following to address his/her further suggestion, "a comment on available data (both in their possession, both published) on analytes' long term stability at -80°C should be added".  Please see Lines 426 to 434.

A flexible dosing will be employed in the clinical trial of EE-TP with the aim of achieving metabolic correction, and thus following dosing and prior to the administration of subsequent doses, a pharmacodynamic evaluation will be necessary.  For this reason, samples will not be subjected to long-term storage prior to analysis. However, to confirm the stability of analytes during the clinical trial sample storage period we are conducting a partial long-term stability validation. We have no reason to believe that there will be a deterioration in metabolite concentrations under the intended storage conditions and for the duration of the clinical trial, as plasma samples for patients 1 and 3 were analysed after a storage period of over three years at -80oC and provided measurements that were equivalent to those that had been measured by a UPLC we previously employed [4, 13].